

# A classical density functional from machine learning and a convolutional neural network

**Shang-Chun Lin[1][*] and Martin Oettel[1]**

**1** Institut für Angewandte Physik, Universität Tübingen,
Auf der Morgenstelle 10, 72076 Tübingen, Germany

[*] shang-chun.lin@uni-tuebingen.de

## Abstract

We use machine learning methods to approximate a classical density functional. The functional 'learns' by comparing the density profile it generates with that of simulations. As a study case, we choose the model problem of a Lennard–Jones fluid in one dimension where there is no exact solution available. After separating the excess free energy functional into a "repulsive" and an "attractive" part, machine learning finds a functional for the attractive part in weighted–density form. The predictions of density profile at a hard wall shows good agreement when subject to thermodynamic conditions beyond those in the training set. This also holds for the equation of state if this is evaluated near the training temperature. We discuss the applicability to problems in higher dimensions.

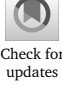

# 1 Introduction

Density functional theory (DFT) for many–body systems is built upon the one–to–one correspondence between the one–body density profile of particles and the one–body external potential acting on these particles [1, 2]. In quantum Kohn–Sham (KS) DFT the particles are electrons with Coulomb interactions and most work has addressed the case of zero temperature ($T = 0$) [3]. In classical DFT, particles can be molecules, macromolecules or colloidal particles with a huge variety of interparticle interactions and DFT addresses the finite temperature case [4]. Both in quantum KS DFT and in classical DFT the theorems of DFT guarantee the existence of a unique functional depending on the density; it is an energy functional for KS DFT and a free energy functional for classical DFT. Densities are computed by self–consistent equations involving functional derivatives of these functionals and these equations are solvable with much less numerical effort than full many–body quantum calculations/simulations or classical simulations. The exact energy/free energy functionals are not known in general, therefore considerable effort has gone into the theoretical development of functionals.

The art of functional building is very different in the quantum and the classical case. In the quantum case, we deal with one type of interparticle interaction (Coulomb interaction) but additionally one has the problem of indistinguishable fermions. Therefore a major problem in quantum DFT is the kinetic energy functional $T[n]$ which is solved largely by introducing KS orbitals (however, at rather high numerical costs). The remaining contributions to the electron energy besides the classical Coulomb energy are hidden in the exchange–correlation functional $E_{xc}[n]$ which in the majority of applications is assumed to be local (the energy density at a given point only depends on densities $n$ and gradients of densities $\nabla n$ at this point). In the classical case, the equivalent of Coulomb energy plus $E_{xc}[n]$ is the excess free energy functional $\mathcal{F}^{ex}[\rho]$ depending on the particle number density $\rho$. Due to the variety of particle interactions in classical system, there is no unique model–building recipe for $\mathcal{F}^{ex}[\rho]$. Very often, interparticle potentials feature steeply repulsive cores and more or less smoothly varying attractive and/or repulsive portions outside this core. The steepness of the potential in general makes local approximations to $\mathcal{F}^{ex}[\rho]$ very unreliable. Considerable progress over the past decades has been made for the case of hard–body fluids where the hard interaction serves as an idealized approximation to the repulsive cores in the interaction of realistic fluids. In particular, excess functionals derived from Fundamental Measure Theory (FMT) [5] have a high degree of accuracy. For anisotropic particles, these are rather recent achievements [6–8]. On the other hand, the contribution to $\mathcal{F}^{ex}[\rho]$ from attractive interactions outside the hard core is treated by mean–field concepts in various guises (random phase approximation (RPA) [9], functional expansions [10, 11], Wertheim theory for patchy attractions [12, 13] etc.) but a qualitatively new and successful *ansatz* (such as FMT was in the treatment of hard bodies) is missing.

With the steep increase of available computing power over the past years, methods of machine learning (ML) have come into the focus of research also in physics. ML is designed for finding patterns in high–dimensional data. Algorithms of ML still rely on insight and intuition how to represent and process data but a detailed model–building (specific for the problem at hand) is not required. The optimization of data representation/processing can be viewed as a numerically intensive data fitting task which is very familiar to physicists. Therefore it appears also natural to apply ML to the problem of functional construction in DFT. In the past years, such ideas have been driven by the quantum DFT community. Refs. [14, 15] address the construction of a ML functional for the kinetic energy functional $T[n]$ in one dimension (1D). Although successful in obtaining energy values, certain limitations when going to three dimensions (3D) have led the authors of Ref. [16] to apply ML directly to the functional map between the external potential and the electron density. Although the approach appears to be quite promising in terms of possible accuracy, it amounts to hiding the energy functional in a

"ML black box", which might appear less appealing to theorists.

Certainly, in the case of a "ML black box" functional one must be careful in choosing training data sets in relation to the applications one has in mind. For classical DFT, training data sets would be created most naturally by Monte Carlo (MC) or Molecular Dynamics (MD) simulations. To keep numerical efforts down, training sets should be created using small sets of parameters (chemical potential, temperature, external potentials) with good statistics. For a classical "ML black box" functional, the highly nonlinear packing effects would probably necessitate to train the ML functional with densities at least as high as in the application cases. On the other hand, packing is well described by the existing hard–body functionals and one may doubt whether the currently existing ML schemes can improve those. Therefore, for a fluid with repulsive cores it makes sense to maintain the splitting of the excess free energy functional into a hard core part and a part describing the soft parts of the potential. In this paper, we consider a Lennard–Jones (LJ) model fluid in 1D (where this soft part is attractive) and we aim to find a ML functional for the attractive part of $\mathcal{F}^{\mathrm{ex}}[\rho]$ while representing the repulsive part by the exactly known hard rod functional.

A LJ model in 1D is not of the nearest–neighbor interaction kind for which exact functionals (but only implicitly known) exist [17,18]. Therefore, training data sets have to be obtained by simulations (similar to desired extensions to 3D). In 1D, mean–field approximations for the attractive part of $\mathcal{F}^{\mathrm{ex}}[\rho]$ suffer from predicting an unphysical vapor–liquid transition. However, a study for a 1D nearest-neighbor fluid showed rather good results for pair correlations as obtained from explicit minimization of a RPA–like functional [9]. For our LJ fluid, the RPA functional performs somewhat worse (see below) and this finding therefore constitutes a case for improving by ML fitting. The ML functional will be constructed using weighted densities which are convolutions of the density with weight functions to be determined by ML fitting. Our ML fitting is similar to a basic generative convolutional neural network which is used in image processing. In convolutional neural networks, input image data are passed through convolution kernels and a nonlinear function (describing "neuron firing") thus obtaining convoluted data. "Supervised" training occurs when image label data ("cat", "dog" etc.) are compared to output labels. These output labels are obtained by further processing the convoluted data by pooling and reduction steps (using so–called perceptrons). "Unsupervised" training would correspond in comparing the input image with an output image generated from the convoluted data (this is the generative step). In our case, input data are MC density profiles, the convolution kernels are the weight functions and the nonlinear function corresponds to the self-consistent minimization equation, generating directly an output density profile. Therefore, the ML functional is obtained by unsupervised training in the language of the ML community.

The remainder of the paper is structured as follows: In Sec.2 we briefly recapitulate the necessary elements of classical DFT and introduce the model in Sec.3. In Sec.4 we describe our results and in Sec.5 we conclude with a summary and a discussion of possible future work.

## 2   Classical DFT

In classical DFT, the grand potential functional is

$$\Omega[\rho(x)] = \mathcal{F}^{\mathrm{id}}[\rho(x)] + \mathcal{F}^{\mathrm{ex}}[\rho(x)] + \int dx (V^{\mathrm{ext}}(x) - \mu)\rho(x), \tag{1}$$

where $\rho(x)$ is particle density distribution, $\mathcal{F}^{\mathrm{id}}$ is free energy functional of the ideal gas, $\mathcal{F}^{\mathrm{ex}}$ is the excess free energy functional from the particle interactions, $\mu$ is chemical potential and

$V^{\text{ext}}$ is the external potential. The exact form of $\mathcal{F}^{\text{id}}$ is:

$$\beta \mathcal{F}^{\text{id}} = \int dx\, \rho(x)[\ln(\rho(x)\lambda) - 1], \tag{2}$$

with $\beta = 1/k_B T$, $T$ the temperature, $k_B$ Boltzmann constant, and $\lambda$ the thermal wavelength. In the following we set $\beta = 1/k_B T = \lambda = 1$. In equilibrium, the density profile $\rho^{\text{eq}}$ must minimize $\Omega$ for a given $\mu$. Thus, with $\frac{\delta \Omega}{\delta \rho} = 0$ and Eq. (2), we obtain

$$\rho^{\text{eq}} = \exp\left(\mu - \left.\frac{\delta \mathcal{F}^{\text{ex}}}{\delta \rho}\right|_{\rho = \rho^{\text{eq}}} - V^{\text{ext}}\right). \tag{3}$$

To solve Eq. (3), a robust but sometimes not very efficient method is Picard iteration with mixing:

$$\rho^{\text{new}}(x) = \exp\left(\mu - \left.\frac{\delta \mathcal{F}^{\text{ex}}}{\delta \rho}\right|_{\rho = \rho^{\text{i}}} - V^{\text{ext}}\right), \tag{4}$$

and the density profile in step $i + 1$ is obtained from step $i$ by $\rho^{\text{i}+1} = (1 - \xi)\rho^{\text{i}} + \xi\rho^{\text{new}}$ with a suitable mixing parameter $\xi$ ($0 < \xi < 1$), until $\rho^{\text{i}} = \rho^{\text{new}}$. All the predictions of density distribution in this paper are initialized by a constant value and iteratively solved using Eq. (4).

In this paper, we investigate a pair potential between particles given by the LJ potential:

$$U_{\text{LJ}}(x) = \begin{cases} \infty & \text{if} \quad x < \sigma \\ 4\epsilon\left[\left(\frac{\sigma}{x}\right)^{12} - \left(\frac{\sigma}{x}\right)^6\right] & \text{if} \quad \sigma < x < 16\sigma \\ 0 & \text{otherwise,} \end{cases}$$

with $\sigma$ the diameter of the particles and $\epsilon$ the strength of interaction. In order to construct the free energy functional, we split $\mathcal{F}^{\text{ex}}$ into a reference system functional $\mathcal{F}^{\text{ref}}$ (describing the effect of the repulsive part in $U_{\text{LJ}}$) and a remainder describing the attractive part. The respective splitting of $U_{\text{LJ}}$ into a repulsive and attractive part is performed via the Weeks-Chandler-Andersen (WCA) prescription [19, 20]

$$U_{\text{rep}}(x) = \begin{cases} U_{\text{LJ}}(x) + \epsilon & \text{if} \quad x < 2^{1/6}\sigma \\ 0 & \text{otherwise} \end{cases}$$

and

$$U_{\text{att}}(x) = \begin{cases} -\epsilon & \text{if} \quad x < 2^{1/6}\sigma \\ U_{\text{LJ}}(x) & \text{otherwise.} \end{cases}$$

Naturally, the hard rod functional $\mathcal{F}^{\text{HR}}$ is chosen for $\mathcal{F}^{\text{ref}}$ [21] (see also appendix). Furthermore, we define the RPA–like mean field (MF) approximation, $\mathcal{F}^{\text{ex}} = \mathcal{F}^{\text{HR}} + \mathcal{F}^{\text{MF}}$ with

$$\mathcal{F}^{\text{MF}} = \frac{1}{2} \int \int \rho(x)\rho(x')U_{\text{att}}(|x - x'|)dx\,dx'. \tag{5}$$

## 3 Machine learning model

In order to construct a ML fitting procedure, we split $\mathcal{F}^{\text{ex}}$ into $\mathcal{F}^{\text{HR}}$ and $\mathcal{F}^{\text{ML}}$. $\mathcal{F}^{\text{HR}}$ is given by Eq. (13) (see appendix) and $\mathcal{F}^{\text{ML}}$ is the remainder functional for the attractive part to be found by ML, improving $\mathcal{F}^{\text{MF}}$. The network will be trained by grand canonical simulation data for

the density profile, $\rho^{\mathrm{MC}}(x)$, for a restricted set of the parameters $\{\mu, \epsilon, V^{\mathrm{ext}}\}$. Using Eq.(3), we define a ML output density as

$$\rho^{\mathrm{ML}}(x) = \exp\left( \mu - \frac{\delta(\mathcal{F}^{\mathrm{HR}} + \mathcal{F}^{\mathrm{ML}})}{\delta\rho}\bigg|_{\rho=\rho^{\mathrm{MC}}} - V^{\mathrm{ext}} \right), \tag{6}$$

which corresponds to the generative step in a generative convolutional network to determine $\mathcal{F}^{\mathrm{ML}}$. Further, the cost function $J$ is defined as

$$J = \sum_{k=1}^{M} \int_0^L \left( \rho_k^{\mathrm{MC}}(x) - \rho_k^{\mathrm{ML}}(x) \right)^2 dx, \tag{7}$$

where $M$ is number of training samples. To minimize $J$, we choose stochastic gradient descent as the back propagation method, details can be found in the appendix.

In Eq.(6), if $\mathcal{F}^{\mathrm{ML}}$ is exact, then it will generate output $\rho^{\mathrm{ML}}$ which is equal to the input $\rho^{\mathrm{MC}}$. The task is to find a suitable *ansatz* of $\mathcal{F}^{\mathrm{ML}}$ that minimizes $J$. For $\mathcal{F}^{\mathrm{ML}}$ we will consider forms which locally depend on weighted (convoluted) densities

$$n_i(x) = \int_{-L_\omega/2}^{L_\omega/2} \rho(x + x')\omega_i(x') \, dx', \tag{8}$$

with weighting kernels $\omega_i$ that have a range (cutoff length) $L_\omega$. The use of weighted densities is motivated by the proven success of weighted-density formulations as in FMT for hard–body interactions. (Note that the MF approximation (5) has a particularly simple weighted density form.) For example, assuming $\mathcal{F}^{\mathrm{ML}}[n] = \int dx \sum_{ij} \beta_{ij} n_i n_j$ (as in one example below), the trainable parameters $\beta_{ij}$ and $\omega_i(x)$ will be tuned in minimizing $J$ (see appendix). Such a minimization process is analogous to a generative convolution network [22] with 5 layers: input layer ($\rho^{\mathrm{MC}}, \epsilon, V^{\mathrm{ext}}$), convolutional layer(weighted densities), fully connected layer ($\mathcal{F}^{\mathrm{ML}}$), generative layer (Eq.(6)), and output layer ($\rho^{\mathrm{ML}}$).

## 4 Results

To prepare training samples, we generate $\sim 100$ density distributions with different $V^{\mathrm{ext}}$ by grand canonical MC simulation. For one density distribution, we use $10^6$ trial moves to equilibrate, and sample $10^8$ times to calculate the histogram of the density distribution with grid spacing $\Delta x = \frac{1}{8}\sigma$. (The same gridding is chosen for the numerical evaluation of the functionals later on). To decorrelate, samples are separated by 1024 trial moves. As training external potentials in a box with $x \in [0, L]$, we choose a set of soft walls with tunable strength, steepness and wall distance:

$$V^{\mathrm{ext}}(x) = \begin{cases} a((L/2 - b\sigma) - x)^c & \text{if} \quad x \leq L/2 - b\sigma \\ a(x - (L/2 + b\sigma))^c & \text{if} \quad x \geq L/2 + b\sigma \\ 0 & \text{otherwise,} \end{cases}$$

with random parameters $a, b, c$, in the range 1...3, 6...14, 2...4, respectively, and fix the system size to $L = 32\sigma$. Examples are shown in Fig.1.

### 4.1 Training at constant temperature and chemical potential ($\epsilon$ and $\mu$ fixed)

Here 64 training density distributions are generated with fixed $\mu = \ln 1.5$ and $\epsilon = 0.5$ (corresponding to a fixed temperature). The cutoff length $L_\omega$ is $6\sigma$. To test the quality of the

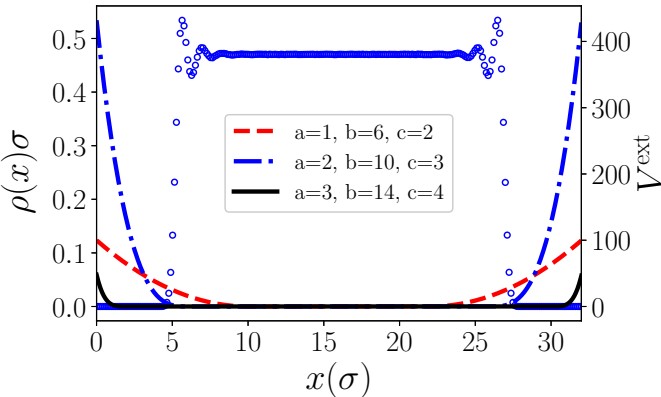

Figure 1: One example of $\rho^{\mathrm{MC}}(x)$ and three examples of $V^{\mathrm{ext}}(x)$. Blue circles shows $\rho^{\mathrm{MC}}(x)$ corresponding to $V^{\mathrm{ext}}(x)$ with $a = 2, b = 10$ and $c = 3$. Lines show $V^{\mathrm{ext}}(x)$ for three different sets of $\{a, b, c\}$.

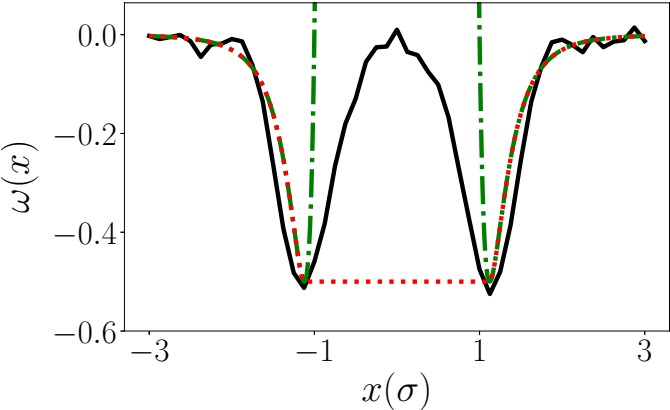

Figure 2: RPA–like mean–field approximation. The black solid line is the ML–optimized $\omega(x)$ appearing in Eq. (9). For comparison $U_{att}/(2\epsilon)(x)$ (red dots) and the full LJ potential $U_{\mathrm{LJ}}/(2\epsilon)(x)$ (green dot–dashed lines) are shown.

extracted $\mathcal{F}^{\mathrm{ML}}$, the pressure $P(\rho)$ (equation of state) and the density $\rho(\mu)$ at the same temperature ($\epsilon = 0.5$) but different chemical potentials compared to the training are compared to MC simulation values. Also, we will test the density distribution in contact with a hard wall, $\rho^{\mathrm{wall}}(x)$, which is equivalent to $V^{\mathrm{ext}}(x) = \infty$ if $x < \sigma$ and 0 otherwise.

### 4.1.1 Learning an improved RPA mean field functional

Consider an extremely simple case,

$$\mathcal{F}^{\mathrm{ML}}_{o=1}[\rho] = \epsilon \int dx\, \rho(x) n(x). \tag{9}$$

This is the weighted–density form of the RPA mean field approximation (5). Thus, the kernel $\omega$, as described in Eq.(8), should correspond to $U_{att}/(2\epsilon)$. Fig. 2 shows the final result of the kernel $\omega$ after training. The tail of $\omega$ is close to $U_{att}/(2\epsilon)$ as expected, and it extends somewhat into the hard core region but goes to zero there ($-\sigma < x < \sigma$), which indicates that the WCA prescription overestimates the attractive potential inside the core. Further, we show the equation of state in Fig.3 and $\rho^{\mathrm{wall}}_{o=1}(x)$ in Fig.4 (green dot–dashed lines). The equation of

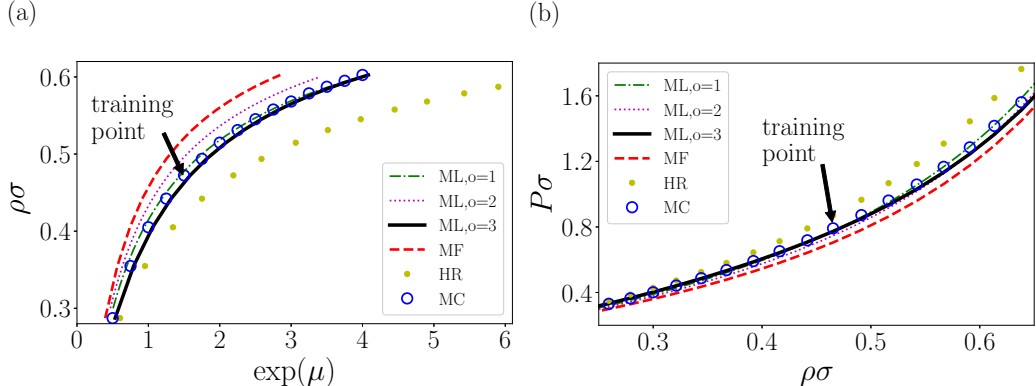

Figure 3: Equation of state: (a) density $\rho(\mu)$ and (b) pressure $P(\rho)$ for $\epsilon = 0.5$. The blue circles are simulation results and lines are results from the ML and MF functionals. Yellow dots correspond to results from the hard rod functional and serve as a reference.

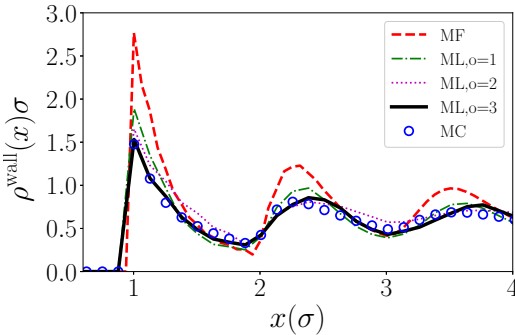

Figure 4: $\rho^{\text{wall}}(x)$ with $\mu = \ln 1.5$ and $\epsilon = 1.5$ (not in training set). The training data are for $\epsilon = 0.5$ and $\mu = \ln 1.5$. The blue circles are simulations and lines are predicted by ML and MF functionals

state is in remarkably good agreement with simulation, but $\rho^{\text{wall}}_{o=1}(x)$ shows strong oscillation and overestimates the density at $x = \sigma$, which is similar to the MF approximation from the WCA separation.

### 4.1.2 Functionals with higher order in $n$

Since the RPA mean field type assumption shows deficiencies in the hard wall profiles, we consider a quadratic form in $n$,

$$\mathcal{F}^{\text{ML}}_{o=2}[\rho] = \epsilon \int dx \sum_{ij} \beta_{ij} n_i n_j, \tag{10}$$

with i, j=0,1...7 (in total eight kernels) and $\beta_{ij} = \beta_{ji}$. Eq. (10) correspondences to a deconvolution *ansatz* for the Mayer $f$–bond $f = \exp^{-U_{\text{att}}} -1$ in the low density and low $\epsilon$ limit. After training, the predicted $\rho^{\text{wall}}_{o=2}(x)$ is shown in Fig.4. Despite the $\rho^{\text{wall}}_{o=2}(x)$ is less oscillatory and seems quite reasonable compared to the simulation, the equation of state does not agree with simulations as shown in Fig.3. The predicted equation of states does not improve with more kernels and longer cutoff length.

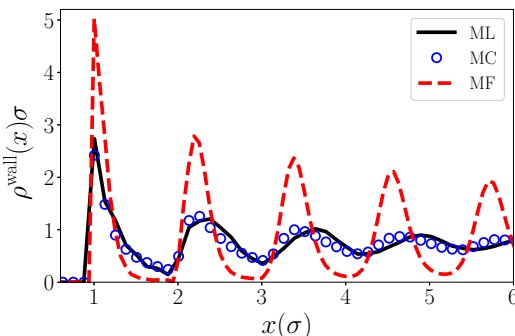

Figure 5: $\rho^{\text{wall}}(x)$ with $\epsilon = 2$ and $\mu = \ln 2$. The blue circles are MC simulations, the black line is the prediction from the ML functional (Eq. (11)) and the red line is determined by the RPA MF approximation (Eq. (5)). The training data are obtained with $\epsilon = 1.0...1.5$.

Thus, we further consider a functional with added cubic term in $n$:

$$\mathcal{F}^{\text{ML}}_{o=3}[\rho] = \epsilon \int dx \left( \sum_{ij} \beta_{ij} n_i n_j + \sum_{ijk} \gamma_{ijk} n'_i n'_j n'_k \right), \tag{11}$$

where $i, j, k$ run from 0 to 7 such that we have in total 16 unknown weight functions. As shown in Fig. 3 and Fig. 4, the equation of state and the predicted $\rho^{\text{wall}}_{o=3}(x)$ (black lines) are now in good agreement with simulation results.

It is worth to note that the information about the exact equation of state is unknown to the network, as we have only one training point for $\mu$. Out of curiosity, we also tried Eq. (11) without the quadratic term $\beta_{ij} n_i n_j$, and it still predicts reasonable density distributions and equation of state.

## 4.2 Training at variable temperature and chemical potential

With the success of the *ansatz* in Eq. (11), we further extend training it to a general training data set with variable $\epsilon$ and $\mu$. Here, 128 training data are generated with random $\mu$ and $\epsilon$ in the range of $\ln 1.5...\ln 3$ and $1.0...1.5$, respectively. The number of kernels is still 16 with range $L_\omega = 6\sigma$ as before. For testing, we use the hard wall density profile $\rho^{\text{wall}}(x)$ for $\mu = \ln 2$ and $\epsilon = 2$, which is not in the training set (it is at a lower temperature, corresponding to larger attractions as in the training sets). The results are shown in Fig. 5 and the predicted $\rho^{\text{ML}}(x)$ is much closer to the simulation than the RPA MF approximation (5).

In Fig. 6, we show the pressure $P(\rho)$ for $\epsilon = 2$ and $\epsilon = 2.5$. The result from $\mathcal{F}^{\text{ML}}$ is in good agreement with our MC simulations for $\epsilon = 2$ while not for $\epsilon = 2.5$. Here we encounter a peculiarity of 1D systems. Even for arbitrarily low temperatures (arbitrarily high $\epsilon$) the pressure $P(\rho)$ must be monotonically increasing, signalling the absence of a phase transition as required for 1D. The RPA mean–field form $\mathcal{F}^{\text{MF}}$ (Eq. (5)) necessarily entails a nonmonotonic $P(\rho)$ (with a van der Waals (vdW) loop) for $\epsilon > \epsilon_c$ where $\epsilon_c$ corresponds to a critical temperature. For $\mathcal{F}^{\text{MF}}$, $\epsilon_c \approx 2.297$ and thus the vdW loop is present for $\epsilon = 2.5$ (see Fig. 6). For $\mathcal{F}^{\text{ML}}$ (Eq. (11)) we find $\epsilon_c \approx 2.989$. It is an improvement that the onset of an unphysical liquid–vapor transition has been moved to higher $\epsilon$ (lower temperatures) but the precursor of it is seen in Fig. 6 for $\epsilon = 2.5$. Thus the failure is understandable since the training data are in the range of $\epsilon = 1.0...1.5$ and Eq.(11) only approximates the functional up to first order of $\epsilon$. Treating higher $\epsilon$ in 1D would require a more sophisticated *ansatz* for $\mathcal{F}^{\text{ML}}$. However, exact results for the equation of state in 1D attractive fluids with strictly next–neighbor interactions

(a)              (b)

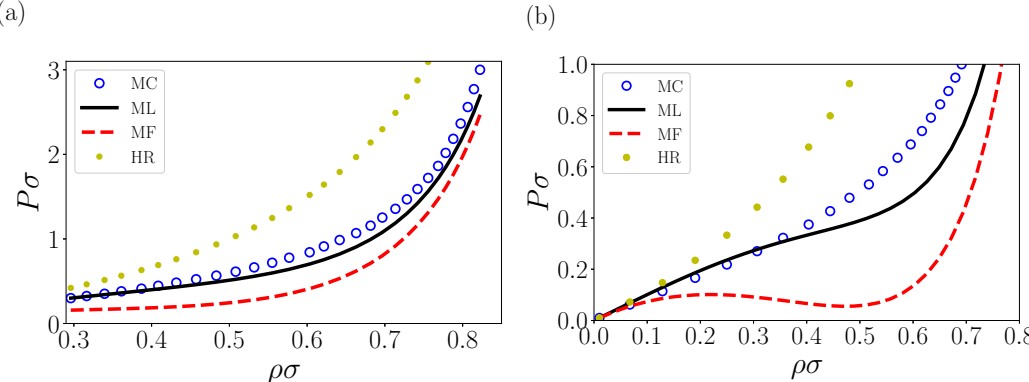

Figure 6: Pressure $P(\rho)$ for (a) $\epsilon = 2$ and (b) $\epsilon = 2.5$. Blue circles are simulations, black lines are bulk results from the ML functional (Eq. (11)) and red dashed lines results from the RPA MF aproximation. For $\epsilon = 2.5$, the RPA MF approximation shows clearly a vdW loop and thus phase coexistence while ML and simulations do not.

(i.e. rather short–ranged attractions) point to a rather complicated dependence on $\epsilon$ [23]. We expect that this problem is absent in a prospective application in 2D or 3D.

## 5 Conclusion

We have introduced a prototype method to obtain a classical density functional for a simple fluid within the framework of unsupervised machine learning, using a method which is akin to a generative, convolutional network. The method is analyzed for the example of a Lennard–Jones fluid in 1D. We have retained the phenomenologically successful splitting of the functional into a reference part (describing repulsive cores and approximated by FMT for hard particles) and a remainder describing attractions. This part is expanded using a set of weighted densities whose functional form and strength is determined by ML. Training data are generated for systems in slits with variable external potentials for a limited range of chemical potentials and temperatures. In evaluating the performance of the ML functional, we computed the equation of state and density profiles at a hard wall and focused on conditions beyond the training set conditions. As a first check, ML finds a RPA–type functional which is clearly superior to the RPA functional derived from the standard WCA separation of the interaction potential. For a ML functional cubic in weighted densities, the results for the hard wall profiles are very good while there are discrepancies to the simulations for low temperatures in the equation of state. This is presumably a 1D artifact.

In our method, the learning of functionals can be viewed as a fitting of weight functions with a certain *ansatz* for the functional. This *ansatz* certainly limits the applicability by construction and is not simply extendable to generate a "ML black box" functional which uses the generic ML algorithms available in the literature for representing the one–to–one map between external potentials and density profiles. This route still awaits exploration. However, we expect that problems in 2D and 3D are amenable to our method and may extend FMT to soft and/or long range potentials, thus constituting a computational continuation of rare theoretical work such as in Ref. [24].

We conclude with some remarks regarding the computational costs. One training sample in this paper takes around 30 minutes on a GPU (graphics processing unit), and the training process takes typically on the order of a week on a single CPU. The bottle neck of the training

is convolution, which GPGPU calculation (general purpose computation on GPU, CUDA [25]) typically gives speedups with one to two orders of magnitude compared to one CPU. Thus, we would expect the training could be done within hours on suitable GPUs. The extension, for example, to 3D LJ particles may take one day [26] to generate one training density distribution, and training may take a week with GPUs. Thus, ML functionals are definitely obtainable within a reasonable time on multiple GPUs.

## Acknowledgements

Helpful discussion with Daniel de las Heras and Miriam D. Klopotek are acknowledged with gratitude.

**Funding information**  S.–C. Lin thanks Landesgraduiertenförderung Baden– Württemberg for financial support. The authors acknowledge support by the High Performance and Cloud Computing Group at the Zentrum für Datenverarbeitung of the University of Tübingen, the state of Baden-Württemberg through bwHPC and the German Research Foundation (DFG) through grant no INST 37/935-1 FUGG.

## A  Functional derivative and gradient descent

Here we discuss some calculation details of forward and backward propagation in minimizing the cost function. We start with the "generative" equation (6) for the $k$-th training data set

$$\rho_k^{\text{ML}}(x) = \exp\left( \mu_k^{\text{ML}} - \frac{\delta(\mathcal{F}^{\text{HR}} + \mathcal{F}^{\text{ML}})}{\delta\rho}\bigg|_{\rho=\rho_k^{\text{MC}}} - V_k^{\text{ext}} \right). \tag{12}$$

Here, we have denoted the chemical potential $\mu = \mu_k$ of the training set by $\mu_k^{\text{ML}}$ which we will allow to vary in the minimization process. Below we specify $\frac{\delta\mathcal{F}^{\text{HR}}}{\delta\rho}$ and $\frac{\delta\mathcal{F}^{\text{ML}}}{\delta\rho}$, needed to determine the rhs of Eq.(12). First, the exact form of $\mathcal{F}^{\text{HR}}$ is [5, 21]

$$\mathcal{F}^{\text{HR}} = \int \phi[n]\,dx = \int -n_0 \ln(1 - n_1)\,dx\,, \tag{13}$$

with $n_i(x) = \int \rho(x')\omega_i^{\text{HR}}(x-x')dx'$ (convolution), where $\omega_1^{\text{HR}}(x) = \theta(\sigma/2 - |x|)$ and $\omega_0^{\text{HR}}(x) = \frac{1}{2}\delta(\sigma/2 - |x|)$, and $\theta$ is the Heaviside function and $\delta$ the Dirac delta function. Thus,

$$\frac{\delta\mathcal{F}^{\text{HR}}}{\delta\rho} = \sum_i \frac{\partial\phi[n]}{\partial n_i} * \omega_i^{\text{HR}}, \tag{14}$$

with $i = \{0, 1\}$ and $*$ denoting convolution.

We illustrate the determination of $\frac{\delta\mathcal{F}^{\text{ML}}}{\delta\rho}$ with the example $\mathcal{F}^{\text{ML}} = \int dx \sum_{ij}\beta_{ij}n_i n_j$:

$$\frac{\delta\mathcal{F}^{\text{ML}}}{\delta\rho} = \sum_{ij}\beta_{ij}\left( n_i \otimes \omega_j + n_j \otimes \omega_i \right), \tag{15}$$

with $\otimes$ denoting cross correlation. (It is worth to note that convolution usually means cross correlation in the ML community.)

It turns out that the minimization process is unstable if the chemical potential in Eq. (12) is fixed at $\mu_k$ (the chemical potential of the MC training set). In order to stabilize the minimization process, we fix it in each application of Eq. (12) by demanding that

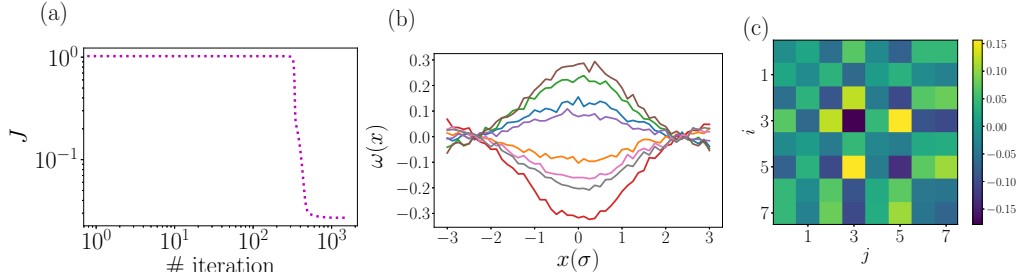

Figure A.1: (a) The cost function $J$ versus number of training iterations and the training result of trainable parameters (b) $\omega(x)$ and (c) $\beta_{ij}$

$\Delta\rho_k = \int (\rho_k^{\text{ML}} - \rho_k^{\text{MC}})^2 dx$ is minimal, which entails that $\mu_k^{\text{ML}}$ varies among different training data sets and during the iterations. If the cost function $J$ converges, $\mu_k^{\text{ML}}$ will converge to a certain number which is not necessarily $\mu_k$. For example, for the training sets with fixed $z = \exp(\mu) = 1.5$ in Sec.4.1, the final averaged $\bar{z}_k = 1.43$ (using Eq. (9)), 1.26 (using Eq. (10)) and 1.56 (using Eq. (11)). The difference between $\bar{z}_k$ and $z$ reflects the residual differences in the equation of state. Both are biggest for the ML functional which is second order in weighted densities (see Fig. 3).

Further, to minimize the cost function $J$ as defined in Eq. (7), we perform stochastic gradient descent, which entails updating the unknown parameters by $\beta_{ij}^{\text{new}} = \beta_{ij}^{\text{old}} - \alpha \frac{\partial J}{\partial \beta_{ij}}$ and the unknown weight functions by $\omega_i^{\text{new}} = \omega_i^{\text{old}} - \alpha \frac{\delta J}{\delta \omega_i}$, where $\alpha$ is the learning rate in the range 0...1 and

$$\frac{\partial J}{\partial \beta_{ij}} = -2 \sum_{k=1}^{M} \int_0^L (\rho_k^{\text{MC}} - \rho_k^{\text{ML}}) \rho_k^{\text{ML}} \left( \frac{\partial \mu_k^{\text{ML}}}{\partial \beta_{ij}} - n_{k,i} \otimes \omega_j - n_{k,j} \otimes \omega_i \right) dx, \qquad (16)$$

where $n_{k,i} = \rho_k^{\text{MC}} \otimes w_i$. The derivative of the chemical potential can be obtained explicitly from the condition $\min(\Delta\rho_k)$:

$$\begin{aligned}
\frac{\partial \mu_k^{\text{ML}}}{\partial \beta_{ij}} = \frac{1}{z_k} \Bigg( & \frac{1}{\int dx (\rho_k'^{\text{ML}})^2} \int dx \rho_k^{\text{MC}} \rho_k'^{\text{ML}} \frac{\partial}{\partial \beta_{ij}} \left( \frac{\delta \mathcal{F}^{\text{ML}}}{\delta \rho} \bigg|_{\rho=\rho_k^{\text{MC}}} \right) \\
& - \frac{\int dx \rho_k^{\text{MC}} \rho_k'^{\text{ML}}}{(\int dx (\rho_k'^{\text{ML}})^2)^2} \int dx \, 2(\rho_k'^{\text{ML}})^2 \frac{\partial}{\partial \beta_{ij}} \left( \frac{\delta \mathcal{F}^{\text{ML}}}{\delta \rho} \bigg|_{\rho=\rho_k^{\text{MC}}} \right) \Bigg),
\end{aligned} \qquad (17)$$

with $z_k = \exp(\mu_k^{\text{ML}}) = \frac{\int dx \rho_k^{\text{MC}} \rho_k'^{\text{ML}}}{\int dx (\rho_k'^{\text{ML}})^2}$ and $\rho_k^{\text{ML}} = z_k \rho_k'^{\text{ML}}$. On the other hand, an approximation can be found by considering a particle reservoir with density $\rho_{0,k}$ at the same chemical potential: $\mu_k^{\text{ML}} = \frac{\delta \mathcal{F}}{\delta \rho} |_{\rho=\rho_{0,k}}$. This gives $\frac{\partial \mu_k^{\text{ML}}}{\partial \beta_{ij}} = n_{k,i}^0 \otimes \omega_j + n_{k,j}^0 \otimes \omega_i$ with $n_{k,i}^0 = \rho_{0,k} \otimes w_i$. The final minimization result is insensitive to the choice of $\frac{\partial \mu_k^{\text{ML}}}{\partial \beta_{ij}}$ and for obtaining the results in this paper we have used the latter expression.

Similarly,

$$\frac{\delta J}{\delta \omega_i(x')} = -8 \left\{ \sum_{k=1}^{M} \int_0^L dx (\rho_k^{\text{MC}} - \rho_k^{\text{ML}}) \rho_k^{\text{ML}} \sum_j \beta_{ij} \left( n_{k,j}^0 - n_{k,j}(x+x') \right) \right\}, \qquad (18)$$

with the assumption $\beta_{ij} = \beta_{ji}$. Here we could see all trainable parameters are coupled with each other.

"Stochastic" gradient descent means in total $N$ training data, only $M$ data are used in one iteration ($N > M$). We have chosen $M = 16$ or 32 in this paper. Starting with $\alpha = 0.01...0.1$,

$\alpha$ is reduced when the cost $J$ increased until $\alpha < 10^{-6}$ and then stopped. As an example, the training results are shown in Fig.A.1 with the training set from Sec.4.1. Also, we tried adding a regularization such as $J' = J + \lambda \left( \sum_{ij} \beta_{ij}^2 + \sum_i \int \omega_i(x)^2 dx \right)$ with $\lambda = 10^{-13} \sim 10^{-14}$, but the effect is negligible. In principle, gradient descent could be applied to arbitrary form of functionals. The minimization procedure has been written in python from scratch, as attempts to use a standard ML library (tensorflow) were not successful.

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
