# Peer review of "A classical density functional from machine learning and a convolutional neural network"

_SciPost Physics, doi:SciPost Phys. 6, 025 (2019)_

## Round 2 · Referee Report · Anonymous (Referee 1) · 2019-1-16

Strengths

The paper presents highly original, creative and modern theoretical work.

Weaknesses

none.

Report

The paper by Lin and Oettel presents a theoretical analysis of a one-dimensional classical system of particles that interact with a generic (Lennard-Jones) model fluid potential. The authors use many-body computer simulations in order to provide quasi exact data on which they base their theoretical considerations. The theoretical aspect of the work spans a broad arch from classical liquid state theory (WCA splitting into repulsive and attractive pair potential contributions) via weighted density DFT to machine learning techniques. In an unexpected turn of events, the authors are able to relate the concept of an ML convolution network specifically to the weight function(s) in a weighted-density approximation (WDA) density functional. Very intriguingly, the authors apply the WDA concept to the *attractive* part of the density functional and obtain the form of the weight functions from ML. Conceptually, this goes far beyond what has been done in electronic structure DFT using ML, where typically brute force learning of the HK functional map from density to external potential is performed. Rather, the hybrid approach of the current paper gives much insight into the formal (analytical) structure of the attractive contribution to the excess free energy functional. Addressing the issue of how to go beyond mean-field attraction has been a long-standing and still open problem for classical fluids. The authors' contribution sheds unexpected light on this very deep issue. Clearly, with the increasing use of ML, this study points towards the future.

I think that this is a brilliant paper, and certainly one of the most inspiring conceptual DFT studies in recent years. I enthusiastically recommend publication of the paper in the present form in SciPost Physics.

Requested changes

none.

  • validity: top
  • significance: top
  • originality: top
  • clarity: top
  • formatting: perfect
  • grammar: perfect

Author:  Lin Shang-Chun  on 2019-02-13  [id 438]

(in reply to Report 1 on 2019-01-16)
Category:
remark

We thank the referee for his/her careful reading of the manuscript, insightful remarks and the very positive evaluation.

---

## Editorial Decision

published